# TRUST BUT VERIFY:
# PROGRAMMATIC VLM EVALUATION IN THE WILD

## ABSTRACT

Vision-Language Models (VLMs) often generate plausible but incorrect responses to visual queries. However, reliably quantifying the effect of such hallucinations in free-form responses to open-ended queries is challenging as it requires visually verifying each claim within the response. We propose Programmatic VLM Evaluation (`PROVE`), a new benchmarking paradigm for evaluating VLM responses to open-ended queries. To construct `PROVE`, we provide a large language model (LLM) with a high-fidelity scene-graph representation constructed from a hyper-detailed image caption, and prompt it to generate diverse question-answer (QA) pairs, as well as programs that can be executed over the scene graph object to *verify* each QA pair. We thus construct a benchmark of 10.5k challenging but visually grounded QA pairs. Next, to evaluate free-form model responses to queries in `PROVE`, we propose a *programmatic* evaluation strategy that measures both the helpfulness and truthfulness of a response within a unified scene graph-based framework. We benchmark the helpfulness-truthfulness trade-offs of a range of VLMs on `PROVE`, finding that very few are in-fact able to achieve a good balance between the two.

## 1 INTRODUCTION

Vision-language models (VLMs) have emerged as an effective solution for generating responses to queries about visual content. However, despite impressive progress (and much like their LLM-counterparts), VLMs are still known to hallucinate – to generate plausible but incorrect answers that are either inconsistent or unverifiable against the provided visual context [1]. This crucial shortcoming has the potential to erode trust in such systems and has already begun to attract significant research (Yu et al., 2024; Liu et al., 2023a; Gunjal et al., 2024; Huang et al., 2024) and regulatory (Biden, 2023) interest, particularly as using such models as the "foundation" of various high-stakes applications becomes imminent (Bommasani et al., 2021).

This has led to a flurry of research on *reliably* benchmarking VLM performance Liu et al. (2024a), by measuring not just the helpfulness but also the *truthfulness* of their responses. Existing benchmarks fall into two categories – *discriminative* (Hu et al., 2023; Lovenia et al., 2023; Li et al., 2023), which evaluate the model's responses to close-ended, existence-based queries ("Is there a man in this image?"), and *generative* (Rohrbach et al., 2018; Sun et al., 2023; Liu et al., 2023b;a; Gunjal et al., 2024), which evaluate responses to free-form, open-ended questions ("Describe this image."). While discriminative benchmarks ease evaluation, they do not realistically simulate in-the-wild usage. On the other hand, generative benchmarks, while realistic, are *extremely* challenging to reliably evaluate, as they require verifying both that the model response fully answers the question (*i.e.* is helpful) and does not make any false claims (*i.e.* is truthful).

Evaluating such free-form responses typically relies on external models (usually, a proprietary LLM) to score responses given some image context (typically ground-truth annotations). However, we find that in several such benchmarks, the context provided is completely insufficient to judge if the response contains hallucinations. Consider Fig. 1: an VLM may respond to the query "How many puppies are in the image?" (correct answer = "four"), with "There are four labradoodle puppies". Evaluating the truthfulness of this statement requires verifying multiple claims about the puppies (`<count == four>` and `<breed == labradoodle>`); however, an LLM judge provided

---

[1]A few LLM-focused works also consider responses that contradict *world knowledge* as hallucinations, but we exclude these from our scope.

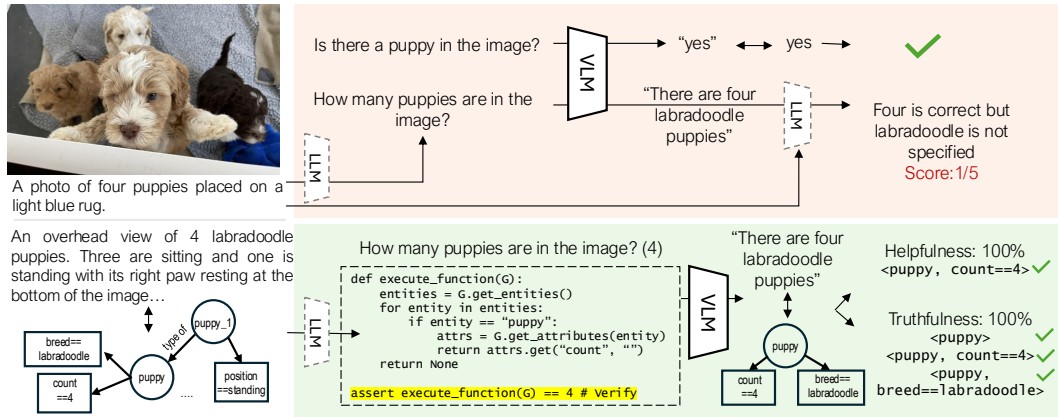

Figure 1: **Top.** Existing VLM benchmarks either limit query-types to easy-to-evaluate but restrictive binary questions, or use external LLMs to generate open-ended questions (without verifying their validity) and score answers (often without complete image context or a clear scoring rubric). **Bottom.** We propose PROVE, a new benchmark that constructs high-fidelity scene-graph representations from hyper-detailed image captions, that are queried via an LLM-generated program to verify a free-form generated question-answer pair. At test-time, we perform an interpretable programmatic evaluation of the helpfulness and truthfulness of free-form VLM responses by comparing scene-graphs.

only with a brief image caption as context ("four puppies placed on a light blue rug") will be unable to do so! Further, the absence of a clear scoring rubric coupled with the sensitivity of LLMs to minor prompt differences often leads to inconsistent and arbitrary scores in such cases. In Fig. 2, we provide real examples from existing benchmarks that illustrate this problem.

We propose Programmatic VLM Evaluation (PROVE), a new evaluation paradigm that performs reliable and interpretable *programmatic* evaluation of free-form VLM responses to challenging, diverse, and grounded questions. To build this dataset, we first use hyper-detailed image captions to construct a high-recall scene graph image representation. We then use an LLM to generate a diverse set of open-ended question-answer (QA) pairs along with accompanying *verification programs*. While the QA pairs are meant to test a range of model capabilities under real-world use, the verification programs can be executed over a given scene graph object to verify the correctness and groundedness of its corresponding QA pair. We thus only retain the QA pairs that we can programmatically verify and construct a benchmark of 10.5k diverse and challenging examples which are visually grounded by design, that we can use to reliably benchmark VLM responses.

Next, we benchmark VLM responses to queries in PROVE by comparing scene graph representations. First, we measure the helpfulness of a response by computing its scene graph-based *recall* against the ground truth answer. Next, we measure response truthfulness as its scene graph-based *precision* against both the scene-graph constructed from the full caption or the image itself. We benchmark a range of VLM responses using this approach, and study their respective trade-offs between helpfulness and truthfulness. Our findings suggest that much of the recent progress in training "better" VLMs also translate to improved helpfulness on our benchmark, but often at the cost of reduced truthfulness.

## 2 RELATED WORK

**Benchmarking VLM hallucination.** Existing benchmarks fall into one of two groups (see Fig. 2):

▷ **Discriminative benchmarks** generate a series of binary questions to verify the presence (or absence) of various entities (or distractors) in the image. Early benchmarks like POPE (Li et al., 2023) limited their scope to object entities annotated by humans or external off-the-shelf models (Zou et al., 2024), , whereas follow-up works additionally evaluate responses to *negative presence* queries (Lovenia et al., 2023), which stress-test the model's abstention capabilities on questions about entities *absent* from the image, or use an LLM to generate a broader range of existence-based questions covering objects and their attributes (Hu et al., 2023). However, while the binary questions that typify such benchmarks simplify evaluation, they do not realistically simulate in-the-wild use.

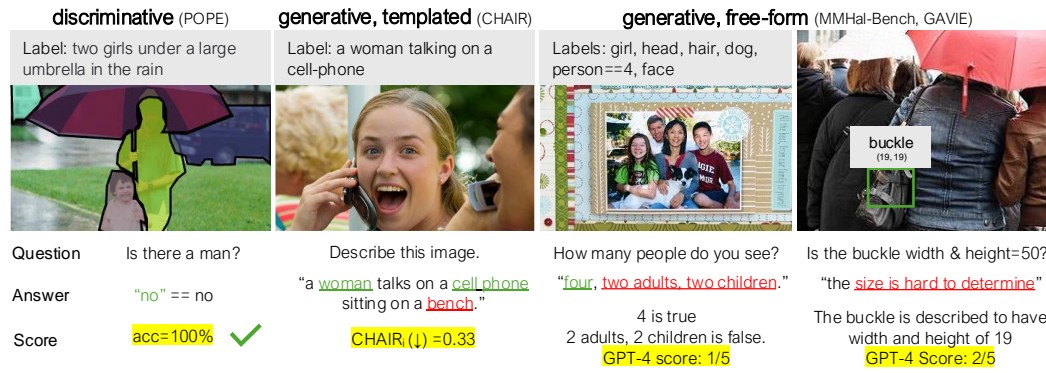

Figure 2: **Top.** Existing VLM hallucination evaluation benchmarks either measure VLM performance on object existence queries ("discriminative" (Li et al., 2023)) or object precision/recall in generated image captions ("generative, templated" (Rohrbach et al., 2018)), neither of which realistically simulate in-the-wild usage. Some recent benchmarks contain open-ended queries ("generative, free-form" (Sun et al., 2023)), which are more realistic but also harder to both generate (*e.g.* see unnatural QA-pair from GAVIE (Liu et al., 2023b) – first from right), and evaluate with an LLM-as-judge (*e.g.* see GPT-4 penalizing a correct response that includes details absent from the ground truth in MMHal-Bench (Sun et al., 2023) – second from right). **Bottom.** We propose PROVE , a benchmark of challenging but verifiable open-ended questions that we use to jointly evaluate both the truthfulness and helpfulness of free-form model responses.

▷ **Generative benchmarks** instead evaluate model hallucinations in response to free-form questions. CHAIR (Rohrbach et al., 2018) measures the precision and recall of entities mentioned in a generated image description against the ground truth. HaELM (Wang et al., 2023b) additionally uses a large language model (LLM) to judge generations, whereas M-HalDetect (Gunjal et al., 2024) has humans annotate hallucinations in model generated descriptions are used to train a predictive model. Recently, AMBER (Wang et al., 2023a) combines a POPE style evaluation with a generative evaluation over an open-ended split. While these benchmarks are indeed more realistic, they still restrict the query instruction to image captioning-style templates ("Describe this image in detail.").

Most recently, a few benchmarks with truly open-ended queries have been proposed (Sun et al., 2023; Liu et al., 2023a; Jing et al., 2023; Liu et al., 2023b), which either hand-design or use an LLM to generate free-form questions, and use external models to judge the corresponding responses. However, these too have limitations: MMHal (Sun et al., 2023) and HallusionBench (Liu et al., 2023a) rely on a series of off-the-shelf models at various stages which introduce noise (see Fig. 2, col 3). GAVIE's (Liu et al., 2023b) reliance on dense captions and bounding boxes leads to a majority of questions querying localized image regions and spatial relationships, many of which have unnatural-sounding responses (*eg.* mentioning image coordinates, see Fig. 2, col 4). Finally, GPT-4-based evaluation is both expensive and inherits the model's own limitations.

**Understanding and mitigating VLM hallucination.** Several works have sought to better understand *why* VLMs hallucinate. One prevalent theory is the model learning spurious correlations between the input and the output: either due to overly strong text priors learned by the LLM backbone (Huang et al., 2024; Leng et al., 2024), or due to distilling synthetic outputs generated by stronger models (such as GPT-4V) that may themselves contain confabulation (Liu et al., 2024b). This is often exacerbated by the predominant training recipe (Liu et al., 2024b; Abdin et al., 2024) that learns a shallow projection from the visual input to the text embedding space which limits the expressivity of the model to learn visually grounded representations.

Recent work has proposed training-based and training-free strategies for mitigating hallucinations. The former involves finetuning (Liu et al., 2023b; Yan et al., 2024) or preference optimization (Yu et al., 2024; Sun et al., 2023) of "preferred" ground truth responses against dis-preferred synthetically generated "hallucinations". Training-free methods instead focus on specialized decoding strategies (Huang et al., 2024; Kim et al., 2024; Leng et al., 2024) that seek to correct for potential statistical bias that may lead to hallucination.

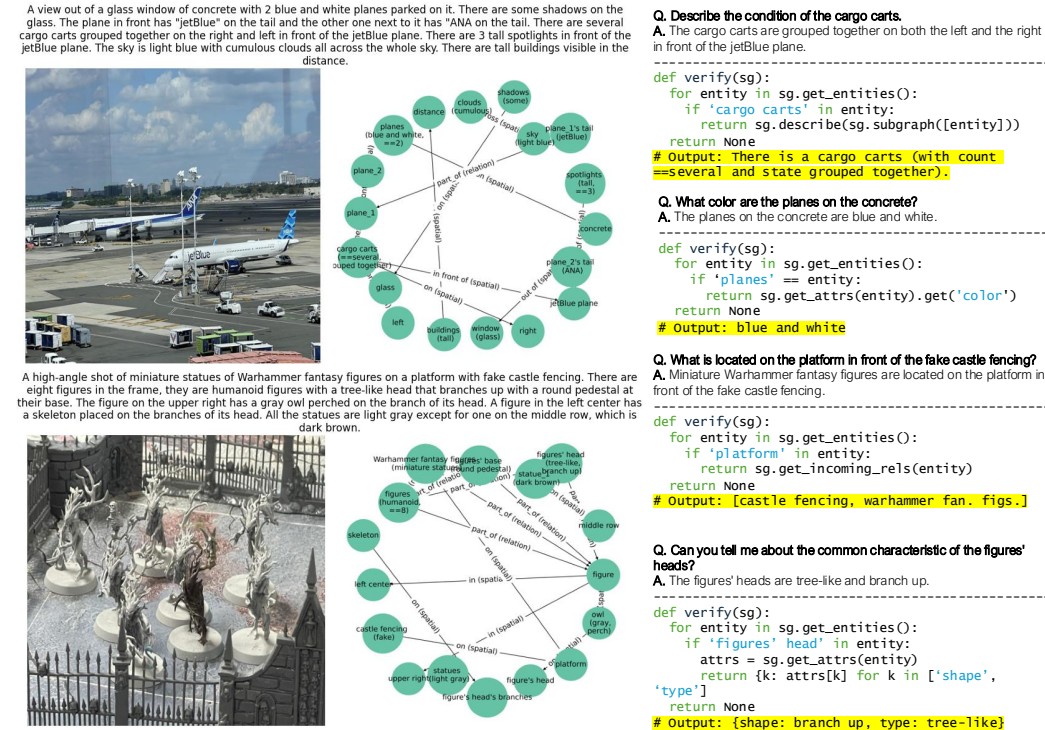

A view out of a glass window of concrete with 2 blue and white planes parked on it. There are some shadows on the glass. The plane in front has "JetBlue" on the tail and the other one next to it has "ANA on the tail. There are several cargo carts grouped together on the right and left in front of the jetBlue plane. There are 3 tall spotlights in front of the jetBlue plane. The sky is light blue with cumulous clouds all across the whole sky. There are tall buildings visible in the distance.

Q. Describe the condition of the cargo carts.
A. The cargo carts are grouped together on both the left and the right in front of the jetBlue plane.
--------------------------------------------------
```
def verify(sg):
    for entity in sg.get_entities():
        if 'cargo carts' in entity:
            return sg.describe(sg.subgraph([entity]))
    return None
# Output: There is a cargo carts (with count
==several and state grouped together).
```

Q. What color are the planes on the concrete?
A. The planes on the concrete are blue and white.
--------------------------------------------------
```
def verify(sg):
    for entity in sg.get_entities():
        if 'planes' == entity:
            return sg.get_attrs(entity).get('color')
    return None
# Output: blue and white
```

A high-angle shot of miniature statues of Warhammer fantasy figures on a platform with fake castle fencing. There are eight figures in the frame, they are humanoid figures with a tree-like head that branches up with a round pedestal at their base. The figure on the upper right has a gray owl perched on the branch of its head. A figure in the left center has a skeleton placed on the branches of its head. All the statues are light gray except for one on the middle row, which is dark brown.

Q. What is located on the platform in front of the fake castle fencing?
A. Miniature Warhammer fantasy figures are located on the platform in front of the fake castle fencing.
--------------------------------------------------
```
def verify(sg):
    for entity in sg.get_entities():
        if 'platform' in entity:
            return sg.get_incoming_rels(entity)
    return None
# Output: [castle fencing, warhammer fan. figs.]
```

Q. Can you tell me about the common characteristic of the figures' heads?
A. The figures' heads are tree-like and branch up.
--------------------------------------------------
```
def verify(sg):
    for entity in sg.get_entities():
        if 'figures' head' in entity:
            attrs = sg.get_attrs(entity)
            return {k: attrs[k] for k in ['shape',
'type']
    return None
# Output: {shape: branch up, type: tree-like}
```

Figure 3: **The PROVE dataset**. For each image-caption pair, we generate a high-fidelity scene graph representation with which we prompt an LLM to generate challenging QA pairs and their verification programs. We only retain QA pairs that we can programmatically verify, ensuring diverse but reliable evaluation data that is grounded by design.

However, developing better understanding and mitigation strategies are both contingent on the availability of reliable evaluation benchmarks. In this work, we introduce such a benchmark with challenging but verifiable open-ended visual questions that we use to jointly evaluate both the truthfulness and helpfulness of free-form model responses.

## 3 APPROACH

Vision-language models are trained to respond to a question $\mathcal{Q}$ about an image $\mathcal{I}$ with a ground-truth answer $\mathcal{A}$. Let $m_\theta(.)$ denote a VLM model trained on a large dataset of such $(\mathcal{I}, \mathcal{Q}, \mathcal{A})$ triplets. At test time, we wish to evaluate the model response $\hat{\mathcal{A}} = m_\theta(\mathcal{Q}, \mathcal{I})$. Specifically, while prior work typically evaluates either the response's correctness (is $\hat{\mathcal{A}} = \mathcal{A}$) or truthfulness (is $p(\hat{\mathcal{A}}|\mathcal{I}) > \text{threshold}$), we propose a unified framework that jointly evaluates both.

### 3.1 GENERATING VERIFIABLE VISUAL QUESTION-ANSWERS

To build PROVE , we first download image-caption pairs $(\mathcal{I}, \mathcal{C})$ from the test set of the recently proposed DOCCI (Onoe et al., 2024) dataset, containing 5k manually curated images with comprehensive human-annotated descriptions. DOCCI is particularly well-suited for VLM evaluation because: i) its captions are extremely detailed, with a higher median caption length than competing datasets, which correlates with high image recall ii) its comprehensive and rigorous 3-stage human annotation protocol leads to high-fidelity captions that are suitable to test a range of image understanding challenges including spatial reasoning, counting, text rendering, and compositionality, and iii) its images are newly curated and so more likely to be truly held-out for existing models.

**Building a robust scene-graph representation.** Scene-graphs are comprised of entity (`<entity>`), attribute (`<entity, attribute>`), and relationship (`<entity_1, attribute, entity_2>`) tuples that describe a scene. We use the tuples included with the DOCCI test set that were automatically extracted from its captions using an LLM (Cho et al., 2024),

and use it to construct a scene graph representation $g(\mathcal{C})$ as a directed graph with attributed entities as nodes and relationships as edges. The scene graph is implemented as a Python class with methods to query the graph for its entities, attributes, and relationships, as well as to extract and describe subgraphs in natural language (full API in Lst. 1).

**Generating open-ended questions with verifiable answers.** Next, for each image, we prompt a pre-trained LLM to generate 10-15 challenging, diverse, and unambiguous question-answer (QA) pairs given a caption and scene graph, along with an accompanying Python program that accepts the scene graph as input and can be executed to verify the generated QA pair (Gupta & Kembhavi, 2023; Surís et al., 2023). We include a few in-context examples of such scene-graph and QA+program input/output pairs in the prompt (see Fig. 9). We repeat this procedure to generate a large dataset of open-ended image+QA pairs $\{(\mathcal{I}_i, \mathcal{Q}_i, \mathcal{A}_i)\}_{i=1}^N$ and their verification programs.

**Filtering QA pairs.** Next, we perform two rounds of filtering:

**1. Programmatic:** First, we execute the generated program with the scene graph as input to verify the QA pair. We discard pairs for which the program either fails or returns an answer that is semantically different (Reimers, 2019) from the ground truth answer.

**2. Text-based:** Next, we perform a few additional post-processing steps to exclude low-quality QA pairs which are i) trivial, ungrammatical, ambiguous, or incomplete (using an LLM, see Fig. 10), ii) not entailed by the image (using a visual entailment model (Wang et al., 2022)), iii) include one or more words from a manually curated list of taboo words that we find to result in low-quality questions, or iv) semantic duplicates for the same image (using SemDeDup (Abbas et al., 2023)). Our final dataset after filtering contains 10.5k high-quality visual question answers – see Fig. 3.

**Dataset statistics.** We now present some statistics about PROVE, which comprises of 10.5k QA pairs generated from 5k image-caption pairs from the DOCCI test set. These are obtained after applying both programmatic filtering *i.e.* either the unit test fails (18.3%) or returns the wrong answer (9.8%), and text-based filtering ($\sim 50\%$ of the total from the previous stage). Note that we opt to filter out such a large percentage of QA pairs in the interest of ensuring high-quality evaluation data. Further, our benchmark curation process is fully automatic and so can be readily scaled to a larger image-caption source. Questions in PROVE average 10.3 words in length whereas answers average 13.4 words (see Fig. 6, *right*). In Fig. 6, *left* we present a sunburst visualization of the first 4 words in the questions, highlighting the diversity of questions in our benchmark.

## 3.2 PROGRAMMATIC VLM EVALUATION (PROVE)

After ensuring the validity of the generated QA pairs, we proceed to evaluating free-form VLM responses to the same $\hat{\mathcal{A}}=m_\theta(\mathcal{Q}, \mathcal{I})$. We first extract tuples from $\hat{\mathcal{A}}$ (using an LLM (Dubey et al., 2024) with in-context prompting), that we use to build a scene graph representation $g(\hat{\mathcal{A}})$. We also build a similar scene graph from the ground truth answer tuples after excluding "premise" tuples included in the question $g(\mathcal{A}) - g(\mathcal{Q})$. We then measure response helpfulness hscore(.) based on *recall* of this scene graph, *i.e.* the fraction of tuples (nodes, attributes, and relationships) in $g(\mathcal{A}) - g(\mathcal{Q})$ that are recovered by $g(\hat{\mathcal{A}})$. Concretely, we compute average cosine similarity between each ground truth tuple and its closest response tuple in embedding (Reimers, 2019) space.

$$\text{hscore}(\hat{\mathcal{A}}) = \frac{\sum_{t \in g(\mathcal{A}) - g(\mathcal{Q})} \max_{t' \in g(\hat{\mathcal{A}})} \text{sim}(t, t')}{|g(\mathcal{A}) - g(\mathcal{Q})|}; \tag{1}$$

Next, we compute tscore(.) as the *precision* of the response *i.e.* the fraction of response tuples that are consistent with either the original scene graph *or* the image itself[2]. We define:

$$\text{tscore}(\hat{\mathcal{A}}) = \frac{\sum_{t' \in g(\hat{\mathcal{A}})} \max\left(\max_{t \in g(\mathcal{C})} \text{sim}(t', t), p(\mathcal{I} \models t')\right)}{|g(\hat{\mathcal{A}})|}; \tag{2}$$

where $\models$ denotes visual entailment, and $p(\mathcal{I} \models t')$ is approximated using a visual entailment model (Wang et al., 2022). Note that hscore and tscore are not necessarily correlated – a response can be helpful (by answering the query) but not entirely truthful (might contain hallucinations), and vice versa. Naturally, different models may achieve different trade-offs between the two – an aspect that PROVE is uniquely suited to analyze.

---

[2]This reduces false-positive hallucination detections, as no caption can capture every aspect of an image.

| Method | #params | hscore (↑) | tscore (↑) | average (↑) |
|---|---|---|---|---|
| Blind Baseline* | 8B | 40.99 | 43.54 | 42.27 |
| Qwen2-VL (Bai et al., 2023) | 2B | 67.53 | 80.89 | 74.21 |
| InternVL2 (Chen et al., 2024) | 2B | **71.97** | 78.97 | 75.47 |
| Phi-3.5-Vision (Abdin et al., 2024) | 4B | 70.21 | **81.79** | **76.00** |
| LLaVA-1.5 (Liu et al., 2024c) | 7B | 70.62 | **82.58** | **76.60** |
| LLaVA-Next (Liu et al., 2024b) | 7B | **72.37** | 79.39 | 75.88 |
| InternVL2 (Chen et al., 2024) | 8B | 71.88 | 79.96 | 75.92 |
| Pixtral (Mistral, 2024) | 12B | 70.74 | 82.04 | 76.39 |
| LLaVA-1.5 (Liu et al., 2024c) | 13B | 71.28 | 82.80 | **77.04** |
| InternVL2 (Chen et al., 2024) | 26B | **73.10** | 79.55 | 76.32 |
| Gemini-1.5-Flash[†] (Team et al., 2023) | - | 69.44 | **81.27** | 75.36 |
| GPT-4o-mini[†] (Achiam et al., 2023) | - | 71.65 | 78.67 | 75.16 |
| Claude-3.5-Sonnet[†] (Anthropic, 2023) | - | 72.57 | 77.06 | 74.81 |
| GPT-4o[†] (Achiam et al., 2023) | - | 74.02 | 80.92 | 77.47 |
| Oracle* | - | 77.19 | 85.15 | 81.17 |

Table 1: Benchmarking VLMs on PROVE (*=LLaMA-3.1 (Dubey et al., 2024) backbone, [†]=closed-source). For each model, we report helpfulness (hscore), truthfulness (tscore), and their average. We find larger and more recent models achieve higher hscore  but not necessarily higher tscore .

## 4 EXPERIMENTS

We now present our benchmarking experiments on PROVE. We include a broad set of models spanning a range of sizes and learning strategies and extensively analyze their performance, including their performance trade-offs. We also conduct a human study to validate both the quality of our benchmark and how well our proposed metrics correlate with human judgement.

### 4.1 SETUP

**Baselines.** We include VLMs of three sizes – small (<5B parameters), medium (5-10B parameters), and large (>10B parameters) – and include both open-source and proprietary models. We also include two additional LLM-based methods – a Blind baseline (is not provided the image), and an "oracle" model as an upper bound (a blind model that is provided with the ground truth caption image). Both LLM-based methods use a LLaMA-3.1-8B backbone (Dubey et al., 2024) with in-context prompting.

**Data.** PROVE  is constructed from images, tuples, and captions released under a CC by 4.0 license as the test split of the DOCCI (Onoe et al., 2024) dataset. DOCCI images were reviewed both by human and automatic methods to remove or obfuscate PII (faces, phone numbers, and URLs) and unsafe content. Images underwent a rigorous 3-stage human annotation phase resulting in hyper-detailed and high-recall captions averaging 136 words.

**Implementation details.** We use GPT-4o (Achiam et al., 2023) for generating structured question, answers, and verification programs using the batch API and prompting it with a detailed task description, examples, and a Python definition of the SceneGraph class. We also use GPT-4o for the first round of text-based post-processing described in Sec. 3.1. We use OFA (Wang et al., 2022) fine-tuned for visual entailment for both post-processing and measuring image-tuple entailment (Eq. 2), and Sentence-Bert  (Reimers, 2019) to extract text embeddings.

### 4.2 RESULTS

Table 1 and Figure 4 present evaluation results. We find that:

▷ **Few models strike a good balance between helpfulness and truthfulness.** As Fig. 4 (left) shows, models tend to exhibit a range of trade-offs between helpfulness and truthfulness, with only one of the models that we study (GPT-4o (Achiam et al., 2023)) managing to strike a good balance between the

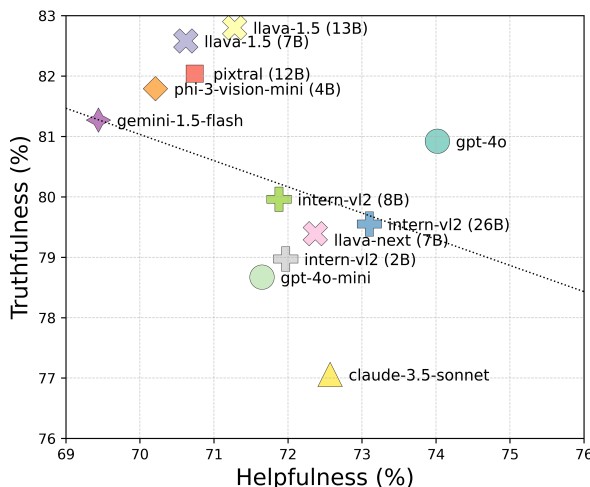

Figure 4: We plot hscore and tscore for VLMs on PROVE – as seen, models with higher helpfulness tend to lag behind on truthfulness, with only GPT-4o striking a good trade-off between the two. Averaged across models, we observe a linear correlation of **-0.43** between hscore and tscore .

two. In fact, we find that many recent models that rank highly on perception and reasoning-focused aggregate benchmarks (Lu et al., 2024), such as Claude-3.5-Sonnet (Anthropic, 2023) and Intern-VL2 (26B) (Chen et al., 2024) tend to indeed achieve higher hscore but lag behind less recent and smaller models like LLaVA-1.5 (Liu et al., 2024c) in tscore. In fact, we find the LLaVA-1.5 model series to obtain the best tscore overall. Overall, we observe a negative linear correlation of **-0.43** between hscore and tscore averaged across models, indicating that atleast some of the recent gains in model helpfulness seem to have been at the cost of lower truthfulness.

Table 1 shows a more detailed breakdown of performance and includes additional baselines. As expected, the blind baseline does significantly worse than others, validating that the image context is indeed crucial for generating helpful and truthful responses. The oracle model, on the other hand, achieves a considerably higher performance than other models, indicating that there is still significant room for improvement in the current generation of VLMs.

▷ **Increasing model size improves** hscore **but not necessarily** tscore. Across both model families that we benchmark at multiple parameter counts and versions – InternVL2 (Chen et al., 2024) (2B, 8B, and 26B), and LLaVA (Liu et al., 2023a) (1.5-7B, Next-7B, and 1.5-13B), we find that larger variants tend to outperform smaller ones in terms of helpfulness but not necessarily truthfulness. Even overall, we find that larger models tend to achieve higher hscore but not necessarily tscore.

▷ **Models fail in different ways.** In Fig. 5 we provide example responses from two models with high hscore (GPT-4o) and tscore (LLaVA-1.5-7B) respectively. We find that while both models struggle with subtasks such as OCR, counting, and reading an analog clock, GPT-4o's errors tend to be less egregious (*e.g.* reading 3/6 letters of the graffiti correctly, while LLaVA only gets 1/6). Further, GPT-4o tends to generate more descriptive answers (*e.g.* correctly identifying that while the wall in the first image is white, the bricks at the bottom are gray), which boost its hscore. In Fig. 7, we include a fine-grained analysis of GPT-4o performance across different question types.

**Human evaluation of PROVE and proposed metrics.** Finally, we conduct two human studies of our benchmark. We first ask human annotators (3 per example) to evaluate the question relevance and answer correctness of QA pairs generated from the qual-test split of the DOCCI dataset (100 images, 170 generated QA pairs) that is specifically set aside for human evaluation. After majority voting, annotators judge 163/170 questions to be relevant (95.9%) and 167/170 answers to be correct (98.2%). We manually inspect the small number of examples judged as irrelevant or incorrect in and find most to be either particularly challenging or subjective, rather than irrelevant or incorrect.

In the second study, we ask subjects (3 per example) to *rate* responses from four models – GPT-4o, LLaVA-1.5-7B, LLaVA-Next-7B, and GPT-4o-mini – on the same set of 170 QA pairs based on their helpfulness (0=unhelpful, 1=helpful) and truthfulness (0=fully false, 0.5=partially false, 1.0=fully

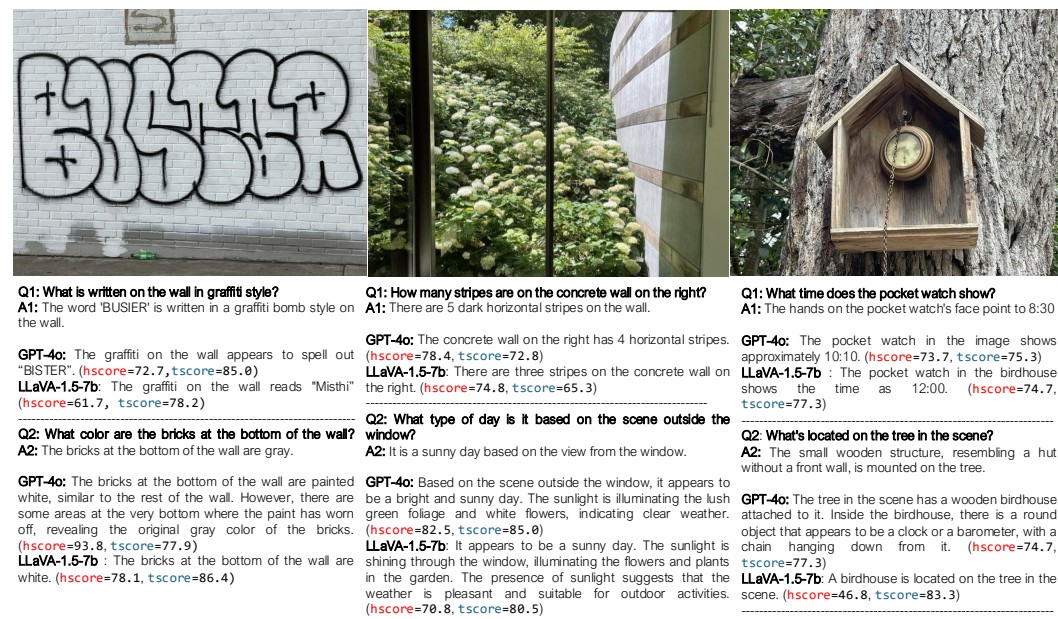

Q1: What is written on the wall in graffiti style?
A1: The word 'BUSIER' is written in a graffiti bomb style on the wall.

GPT-4o: The graffiti on the wall appears to spell out "BISTER". (hscore=72.7, tscore=85.0)
LLaVA-1.5-7b: The graffiti on the wall reads "Misthi" (hscore=61.7, tscore=78.2)

Q2: What color are the bricks at the bottom of the wall?
A2: The bricks at the bottom of the wall are gray.

GPT-4o: The bricks at the bottom of the wall are painted white, similar to the rest of the wall. However, there are some areas at the very bottom where the paint has worn off, revealing the original gray color of the bricks. (hscore=93.8, tscore=77.9)
LLaVA-1.5-7b: The bricks at the bottom of the wall are white. (hscore=78.1, tscore=86.4)

------------------------------------------------------------

Q1: How many stripes are on the concrete wall on the right?
A1: There are 5 dark horizontal stripes on the wall.

GPT-4o: The concrete wall on the right has 4 horizontal stripes. (hscore=78.4, tscore=72.8)
LLaVA-1.5-7b: There are three stripes on the concrete wall on the right. (hscore=74.8, tscore=65.3)

------------------------------------------------------------

Q2: What type of day is it based on the scene outside the window?
A2: It is a sunny day based on the view from the window.

GPT-4o: Based on the scene outside the window, it appears to be a bright and sunny day. The sunlight is illuminating the lush green foliage and white flowers, indicating clear weather. (hscore=82.5, tscore=85.0)
LLaVA-1.5-7b: It appears to be a sunny day. The sunlight is shining through the window, illuminating the flowers and plants in the garden. The presence of sunlight suggests that the weather is pleasant and suitable for outdoor activities. (hscore=70.8, tscore=80.5)

Q1: What time does the pocket watch show?
A1: The hands on the pocket watch's face point to 8:30

GPT-4o: The pocket watch in the image shows approximately 10:10. (hscore=73.7, tscore=75.3)
LLaVA-1.5-7b: The pocket watch in the birdhouse shows the time as 12:00. (hscore=74.7, tscore=77.3)

------------------------------------------------------------

Q2: What's located on the tree in the scene?
A2: The small wooden structure, resembling a hut without a front wall, is mounted on the tree.

GPT-4o: The tree in the scene has a wooden birdhouse attached to it. Inside the birdhouse, there is a round object that appears to be a clock or a barometer, with a chain hanging down from it. (hscore=74.7, tscore=77.3)
LLaVA-1.5-7b: A birdhouse is located on the tree in the scene. (hscore=46.8, tscore=83.3)

------------------------------------------------------------

Figure 5: Example responses from two VLMs that achieve high hscore (GPT-4o) and tscore (LLaVA-1.5 (7B)) respectively. While both models struggle with sub-tasks such as OCR, counting, and reading an analog clock, GPT-4o's errors tend to be less egregious which leads to a higher hscore.

true), and average. We then automatically compute hscore and tscore for the same set of responses and measure the Pearson correlation between the two, observing a moderately strong correlation of 0.54 for hscore and 0.71 for tscore, suggesting that these metrics indeed capture human judgement.

## 5 DISCUSSION

Our work takes a step towards reliably evaluating the helpfulness-truthfulness trade-offs of vision-language models. Our design leverages an LLM prompted with a robust scene graph representation and API to construct "in-the-wild" visual question answer pairs that are *grounded by design*. Further, these QA pairs lend themselves to programmatic evaluation via comparing scene-graph representations. The reliability of our benchmark comes from three factors: i) high-recall human-annotations image captions that seed the scene graphs, which make it possible to (almost) exhaustively validate the veracity of any claim ii) programmatic verification of the generated QA pairs that ensure that both the question and answer are indeed grounded in the visual input, and iii) evaluation metrics that are both holistic (*i.e.* consider all the provided context) and interpretable (*i.e.* provide a concrete scoring rubric based on scene graph-based matching). We will fully open-source our benchmark code and data for research use.

**Limitations.** While we hope that PROVE will serve as a useful test-bed for reliable VLM evaluation and spur future research on the topic, it is not without limitations. While we try to ensure high precision in QA pairs retained in our benchmark (via programmatic verification), this naturally comes at some cost to recall (*i.e.* some hard-to-verify question types may be excluded from the benchmark). Next, even high-recall image captions may not capture every aspect of an image, and so our evaluation may not be able to catch all model hallucinations. Further, our evaluation relies on off-the-shelf models for computing text-embeddings, scene graph tuples, and image-text entailment, and so almost certainly inherits some of their own limitations. Finally, we hope future work will study the effectiveness of recent fine-tuning (Liu et al., 2023b), preference-tuning (Yu et al., 2024; Sun et al., 2023), and training-free (Huang et al., 2024; Kim et al., 2024; Leng et al., 2024) hallucination mitigation strategies on PROVE, as well as agentic models that can plan (Surís et al., 2023; Gupta & Kembhavi, 2023), reason, and self-reflect (Valmeekam et al., 2024), towards the elusive goal of achieving Pareto improvements in both helpfulness and truthfulness.

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

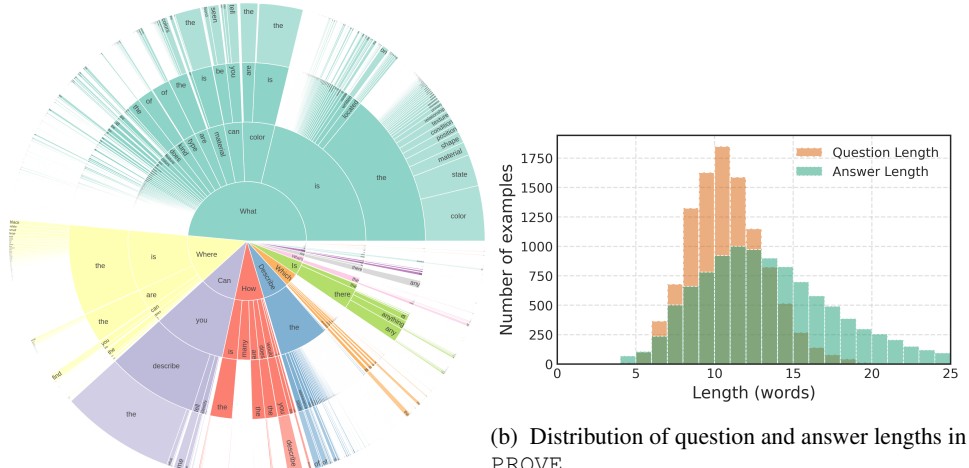

(a) Sunburst visualization of the first 4 question words in `PROVE`.

(b) Distribution of question and answer lengths in `PROVE`.

Figure 6: `PROVE`: Additional dataset statistics.

# A APPENDIX

## A.1 ADDITIONAL DATASET DETAILS

Fig. 6a, *left* presents a sunburst visualization of the first 4 words in the questions within the `PROVE` dataset. As seen, the questions are diverse and span a wide range of question types. Further, while nearly 50% of the questions begin with "What", even this subset spans a range of topics testing numerous model capabilities – see Fig. 5. Fig. 6, *right* shows the distribution of question and answer lengths in `PROVE`. Questions in the dataset average 10.3 words in length, whereas answers average 13.4 words, with both following a normal distribution spread.

## A.2 ADDITIONAL IMPLEMENTATION DETAILS

Lst. 1 provides a Python implementation of the SceneGraph class used to represent scene graphs in `PROVE`. The class provides methods to generate subgraphs, describe subgraphs in natural language, and query entities, attributes, and relationships. We include the prompts used for generating verifiable question-answer pairs as well as the post-processing prompt used to filter out low-quality QA pairs in Figs. 9- 10.

## A.3 ADDITIONAL PERFORMANCE ANALYSIS

Fig. 7 presents a fine-grained performance analysis of GPT-4o on `PROVE`. We break down helpfulness and truthfulness scores by question type, and display the top-10 most common question types sorted by performance. As seen, the model performs particularly well on questions that require reasoning about spatial relationships (where are/is), object attributes (what color), and generating image descriptions.

Fig. 8 shows a word cloud of the most commonly hallucinated objects in answers to questions from `PROVE` across all models. As seen, models commonly hallucinate common objects such as "tree", "building', "wall", and "sign".

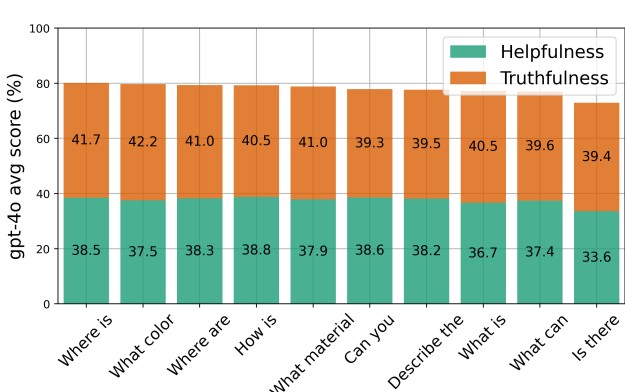

Figure 7: GPT-4o fine-grained performance analysis

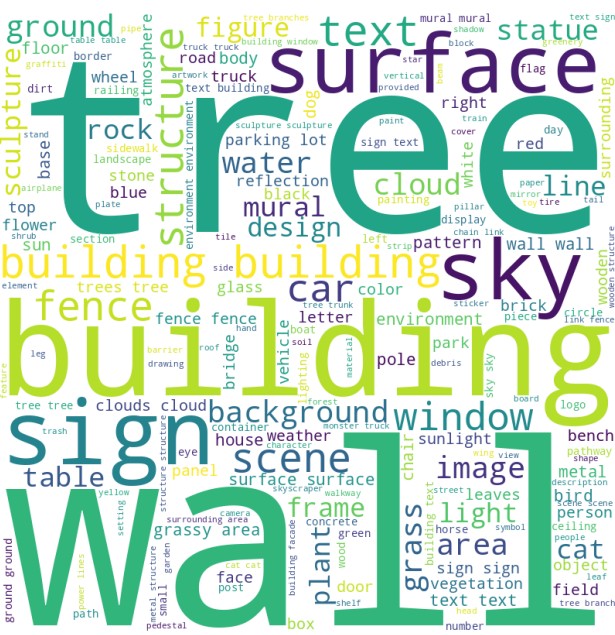

Figure 8: Word cloud of hallucinated objects in model-generated responses.

```python
############################################################
# SceneGraph class API
############################################################
class SceneGraph(nx.DiGraph):
    def __init__(self, caption, sg_dict, *args, **kwargs):
        """Init scene graph from entity-attribute-relationship dict"""
        super().__init__(*args, **kwargs)
        for source_ent, metadata in sg_dict.items():
            self.add_node(source_ent, **metadata["attributes"])
            for target_ent, rel_info in metadata["relations_to"].items():
                self.add_edge(source_ent, target_ent, **rel_info)
        self.caption = caption
        self.sg_dict = sg_dict

    def generate_subgraph(self, node_list)->"SceneGraph":
        """Generates a subgraph with nodes in node_list"""
        return nx.subgraph(self, node_list)

    def describe(self, subgraph): -> str:
        """Generate a natural language description of a subgraph"""
        return generate_description(subgraph)

    def get_entities(self) -> List[str]:
        """Returns a list of entities in the scene graph."""
        return list(self.nodes)

    def get_attributes(self, ent_name) -> dict[List]:
        """
        Returns a list of attributes for ent_name in the scene graph
        Format: { "att_type": f"att_value_1,  att_value_2, ..." }
        """
        return self.nodes.get(ent_name, {})

    def get_outgoing_relations(self, ent_name) -> dict:
        """
        Returns a dict of relations for which ent_name is the source.
        Format:  {target_ent_1: { rel_type_1: [rel_val_1, ...]} ... }
        """
        out_edges = list(self.out_edges(ent_name, data=True))
        out_edges = { tup[1]: {**tup[2]} for tup in out_edges }
        return out_edges

    def get_incoming_relations(self, ent_name) -> dict:
        """
        Returns a dict of relations for which ent_name is the target
        Format: {source_ent_1: { rel_type_1: [rel_val_1, ...]} ...}
        """
        in_edges = list(self.in_edges(ent_name, data=True))
        in_edges =  { tup[0]: {**tup[2]} for tup in in_edges }
        return in_edges
```

Listing 1: Python API for the SceneGraph class

Consider the SceneGraph class defined below, which takes as input an image caption and a dictionary of tuples (entities, attributes, and relations) parsed from the image caption, and builds a directed graph representation with attributed entities as nodes and relations as edges.

You will be provided with such an image caption, tuple dictionary, and corresponding SceneGraph object. Your task is to generate a set of:

1. **Free-form question-answer pairs** that test non-trivial image understanding and reasoning capabilities.

2. **A Python function** that receives as input the SceneGraph object and can be executed to answer the query by reasoning over the scene graph.

**Guidelines.** The generated questions should be:

- **Clear and conversational**, in the tone of a person who is asking another person about the scene. You may paraphrase where appropriate to improve clarity (eg. "Can you describe the dog?" is better than "What is the state of the dog?"). Note that the tuples in the scene graph are generally accurate but not necessarily precise, and so may require rephrasing to generate meaningful questions from.

- **Diverse**, both in question type (*e.g.* starting with "is", "where", "what", "when", "how", "which", "why", etc.) and length.

- **Non-trivial** (eg. avoid "What color are the green trees?") and **unambiguous** (eg. avoid "What is the color of the puppy?" for an image with multiple puppies).

The generated Python functions should be:

- **Executable**: The code should run without requiring modifications.

- **General**: The code should generalize to similar scene graphs as the one provided. Do NOT hard-code specific attributes or relations.

For each image, generate 10-15 such question-answer pairs and corresponding Python functions.

Figure 9: LLM prompt for generating visual question-answer pairs along-with verification programs.

You will be provided with a list of question-answer pairs about an image. Your task is to identify whether each pair has any of the following issues:

1. **Trivial question**. A trivial question can be answered directly from information provided in the question or using common-sense, without requiring looking at the image. Examples:

   **Question**: What is the material of the stadium's horizontal concrete bar?
   **Answer**: The stadium's horizontal concrete bar is made of concrete.
   **Judgement**: Trivial (The question already mentions that the bar is made of concrete)

   **Question**: What text rendering is found on the stop sign?
   **Answer**: The stop sign has white text rendering of the word "STOP".
   **Judgement**: Trivial (Stop signs almost always have the word "STOP" on them)

   **Question**: What feature of the scene reflects sunlight?
   **Answer**: The hard surfaces reflect sunlight.
   **Judgement**: Trivial (Hard surfaces are known to reflect sunlight)

2. **Incomplete answer.** An incomplete answer does not completely answer the question. It may be missing key details or may not provide a full description, or may also be entirely irrelevant. Examples:

   **Question**: What is between the red neon light and the frame?
   **Answer**: The red neon light is behind the metal construction frame.
   **Judgement**: Incomplete (does not answer the question)

   **Question**: How would you describe the trees surrounding the green lake?
   **Answer**: The trees surrounding the green lake are large in size.
   **Judgement**: Incomplete ("large" is not a sufficiently detailed description)

   **Question**: In which part of the image is there no visible cloud coverage?
   **Answer**: The rest of the image has the clear blue sky with no visible cloud coverage.
   **Judgement**: Incomplete ("the rest of the image" is meaningless without context)

3. **Unnatural-sounding**. The question-answer pair may sound awkward, ambiguous, or unnatural. This could be due to its phrasing, structure, or grammar. Examples:

   **Question**: Can you describe the role of the stones in relation to the anemones?
   **Answer**: The stones are covered with anemones and line the bottom of the tank and go up its left side.
   **Judgement**: Unnatural ("role" of stones is odd phrasing and the overall question is ambiguous)

   **Question**: What is the overall shape of the section of grass?
   **Answer**: The section of grass is small in shape.
   **Judgement**: Unnatural ("small" is not a shape)

   **Question**: What kind of state is the sign experiencing due to the brightness?
   **Answer**: Due to the brightness, the sign is experiencing a duller state.
   **Judgement**: Unnatural ("experiencing a state" is awkward phrasing)

Figure 10: LLM text-based post-processing prompt.

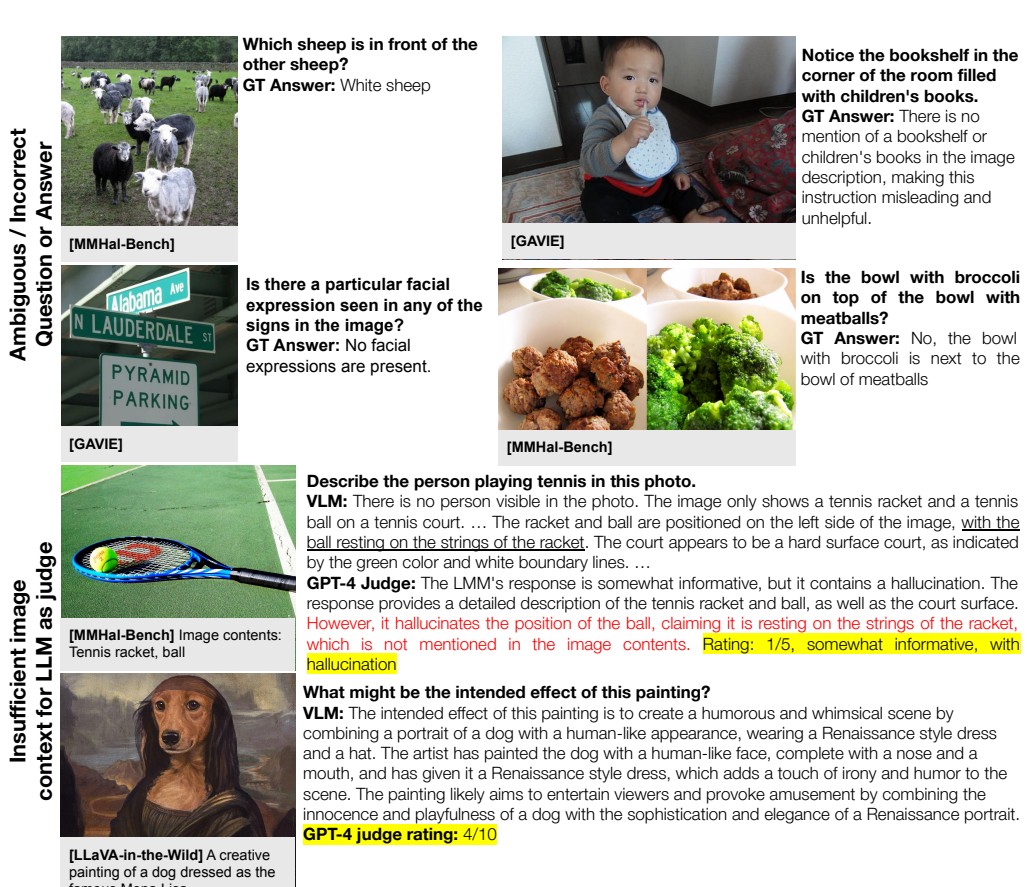

Figure 11: **Failure modes of existing VLM Evaluation Benchmarks.** Existing free-form VLM evaluation benchmarks Liu et al. (2023b); Sun et al. (2023); Liu et al. (2024b) frequently contain ambiguous, incorrect, or unanswerable questions, often due to LLM generation without additional verification. Further, these benchmarks typically rely on pure LLM-as-judge evaluation of free-form VLM responses at test-time, which often leads to unreliable scoring due to the LLM judge having insufficient image context or arbitrary scoring due to the lack of a scoring rubric.

