# OpenReview forum: "Trust but Verify: Programmatic VLM Evaluation in the Wild"
_ICLR.cc/2025/Conference — Submitted to ICLR 2025_

### Official Review · Reviewer_42b9 · 2024-10-26

**Soundness:** 2
**Presentation:** 2
**Contribution:** 2
**Rating:** 5
**Confidence:** 4

**Summary:**

The authors propose PROVE, a new evaluation benchmark for VLMs’ hallucination. PROVE is built on top of the DSG scene graphs of DOCCI dataset, by generating question-answer pairs from the scene graphs and pragmatic/text-based filtering of wrong question-answer pairs. The question/GT answer/verification programs are generated with GPT4o, and the answers generated by VLMs are evaluated with Sentence-BERT and OFA. PROVE has two metrics — hscore (helpfulness) and tscore (truthfulness), based on recall and precision with respect to scene graphs, respectively. The authors evaluate different VLMs in PROVE, and show that different models have different balance between helpfulness/truthfulness. The authors also provide human evaluation showing that PROVE questions are mostly relevant and correct.

**Strengths:**

- (1) Introduction of new hallucination benchmark for VLM based on programmatic questions generated from scene graph.

**Weaknesses:**

- (1) **Methodological contribution is weak.** The authors generate question-answer pairs from the existing Davidsonian scene graphs (DSG; Cho et al., 2024) already shared by DOCCI (Onoe et al., 2024) authors. DSG paper already provides a question-answering generation pipeline from the scene graphs, and it is not clear how the proposed question-answering generation pipeline is more novel or more helpful.

- (2) **No comparison with existing metrics/benchmarks.** The authors mention many VLM hallucination benchmarks in the related work but do not show qualitatively or quantitatively how PROVE is better than existing benchmarks.

**Questions:**

- The citation for OFA seems to be wrong; it should be Wang et al., https://arxiv.org/abs/2202.03052

---

> ### Author Response · Authors · 2024-11-21
> **Author response (1/2)**
>
> [**Methodological contribution is weak .. DSG paper already provides a question-answering generation pipeline from the scene graphs**] We respectfully disagree. Firstly, the goal of DSG is to evaluate *text-to-image* generation.  It does so by generating templated *binary (yes/no) QA pairs*, of the format: _Q: “Is there a <node>”? A: “Yes”_, for every node in the scene graph constructed from the user prompt. The DSG score for a generated image is then computed as the fraction of the scene graph that can be externally verified (e.g. by a VLM) against the generated image.
>
> On the other hand, PROVE seeks to evaluate *image-to-text* generation *in the wild*. As Figure 2 of our paper shows, realistically simulating in-the-wild usage requires evaluating VLM responses to _free-form_ QA pairs that test a range of perception and reasoning abilities, rather than just binary questions. However, naively using an LLM to generate QA pairs at scale, as done in some prior works [A, B], often results in unreliable QA pairs that may be ambiguous, unanswerable, or impossible to reliably evaluate. We include more such examples from existing benchmarks in Figure 11 of the revised appendix.
>
> **Our main methodological contribution** is to address this shortcoming by **generating verification programs** to detect and exclude invalid QA pairs. Note that programmatic verification of free-form QA pairs is far from trivial, as each QA pair requires a bespoke program (see https://prove-explorer-anon.netlify.app for examples). We achieve this by designing a rich scene-graph representation and minimal but highly expressive API to query the scene graph. We then prompt an LLM planner to generate a unique _verification program_ using this API, which is when executed with the scene graph as input can be used to validate its corresponding QA pair. In fact, **28.1%** of the original set of LLM-generated QA pairs (~8100 QA pairs!) are found to be unverifiable and filtered out by this method. This results in a high-quality and reliable evaluation dataset, as confirmed by our human study (L369) where subjects rate 95.9% of questions as relevant and answerable, and 98.2% of answers as correct.
>
> To our knowledge, we are the first to propose such a programmatic verification strategy for validating LLM-generated visual question-answer pairs. Further, we also leverage this scene graph design to propose a novel method for  jointly evaluating the helpfulness and truthfulness of free-form VLM responses at test-time, which leads to more interpretable and trustworthy evaluation than existing LLM-as-a-judge methods (see Figure 11 of the appendix).
>
> [**Comparison with existing metrics**] As suggested, we compare our proposed hscore and tscore metrics to scores generated by an LLM judge (a LLaMA-3.1-Instruct (8b) model) that is provided the ground truth caption as context. We reuse the detailed prompt format introduced in MMHal-Bench [A] that includes several in-context examples, and ask the model to provide scores for both helpfulness and truthfulness. We then repeat the procedure described in L376-405 of our paper: we score responses from four models – LLaVA-1.5 (7b), LLaVA-Next (7b), GPT-4o-mini, and GPT-4o – via this method, and then measure how well these scores correlate with human judgements of the same set of responses.
>
> Shown below are the Pearson correlations between each method (LLM-as-judge and Ours) with human judgements of helpfulness and truthfulness.
>
> |                     |   helpfulness	  |  truthfulness |
> |:-----------------|------------------:|-------------------:|
> | LLM-as-judge | 0.41	          |	0.20		|
> | Ours             |  **0.81** 	|	**0.45**|
>
> As seen, our proposed metrics correlation _substantially_ better with human judgment compared to a pure LLM-as-judge approach. We will include this discussion in the paper, thanks!
>
> [A] Sun, Zhiqing, et al. "Aligning large multimodal models with factually augmented rlhf." arXiv preprint arXiv:2309.14525 (2023).
>
> [B] Liu, Fuxiao, et al. "Mitigating hallucination in large multi-modal models via robust instruction tuning." The Twelfth International Conference on Learning Representations. 2023.

---

> > ### Author Response · Authors · 2024-11-21
> > **Author response (2/2)**
> >
> > [**Comparison with existing benchmarks**] As recommended, we include additional qualitative examples of the failure modes of existing benchmarks in Figure 11 of the appendix. We also include below a table illustrating the main conceptual differences between PROVE and existing benchmarks.
> >
> > |                  | 	Question types | Data source | QA generation | Size  |   Metric |
> > |:-----------------|:------------------|:-------------------|:---------------------|:-----------|:----------|
> > | POPE     |  Binary |	MS-COCO	|  Templated		|3000		| F1 score       |
> > | AMBER |  Binary, Captioning | MS-COCO +Unsplash | Templated |15202	| CHAIR, F1 score|
> > | MMHal-Bench    |    Free-form       	|  OpenImages | Manual |	96	| Judge LLM score     |
> > | GAVIE      |Free-form | Visual Genome | LLM |1000	|  Judge LLM score  |
> > | PROVE (Ours)   | Free-form| DOCCI | LLM + verification	|10500		|  hscore, tscore  |
> >
> > As seen, existing benchmarks either have restrictive question templates (POPE, AMBER) or limited scale due to reliance on human curation (MMHal-Bench). Recent work (GAVIE) generates free-form QA pairs at scale using an LLM prompted with scene graph information, but does not verify the generated QA pairs. Further, existing free-form benchmarks (GAVIE, MMHal-Bench) use pure LLM-as-judge evaluation. PROVE performs additional programmatic verification of LLM-generated QA pairs to improve reliability, and eschews LLM-as-judge evaluation in favor of an interpretable scene-graph based evaluation strategy.
> >
> > Finally, we note that PROVE is a *general* evaluation paradigm for generating challenging but verifiable visual QA pairs. While we choose to construct PROVE from the DOCCI dataset in our paper, it can also be readily applied to other datasets with rich text annotations e.g. Visual Genome [C], BLIP-KALE [D] etc.
> >
> > [**Incorrect citation**] Thanks for catching this, we have corrected it.
> >
> > We sincerely hope that these clarifications persuade you to reconsider your evaluation, and would be happy to address any lingering concerns.
> >
> >
> > [C] Krishna, Ranjay, et al. "Visual genome: Connecting language and vision using crowdsourced dense image annotations." International journal of computer vision 123 (2017): 32-73.
> >
> > [D] Awadalla, Anas, et al. "BLIP3-KALE: Knowledge Augmented Large-Scale Dense Captions." arXiv preprint arXiv:2411.07461 (2024)

---

> > > ### Comment · Reviewer_42b9 · 2024-11-25
> > >
> > > Thanks to the authors for clarifications and additional experiments. I have raised my score from 3 to 5 accordingly.
> > >
> > > I still strongly suggest showing explicit comparison (both qualitatively/quantitatively) between the original DSG framework and the proposed PROVE framework for the _image-to-text generation in the wild_. While the authors' explanation sounds plausible, it's not clear how the proposed modification on top of DSG is effective and necessary for the evaluation of _image-text generation in the wild_ in the current version.

---

> > > > ### Author Response · Authors · 2024-11-25
> > > > **Official comment by authors**
> > > >
> > > > Thank you for considering our response. We are gratified to hear that it has improved your opinion of our work.
> > > >
> > > > As suggested, we include additional comparisons between the original DSG framework and PROVE.
> > > >
> > > > We first tabulate the methodological differences between DSG and PROVE with respect to image-to-text generation in the wild.
> > > >
> > > > |                      | 	Question types | Question generator  | Answer generator | Mean answer length (words) | Metrics |
> > > > |:-----------------|:------------------|:-------------------|:---------------------|:-----------|:----------|
> > > > | DSG     |  Binary |	Template (_Is there a <node>?_)	|  Trivial (always _"Yes"_)		|	1 	|  Accuracy |
> > > > | PROVE |  Free-form | LLM | LLM + Programmatic verification | 13.4  | hscore, tscore|
> > > >
> > > > PROVE seeks to _simulate_ in the wild usage wherein an VLM may encounter open-ended queries that require a diverse range of perception and reasoning skills and/or a descriptive free-form answer. Beyond realism, we argue that restricting question types can also lead to misleading conclusions. To demonstrate, we measure the truthfulness of responses from two models (GPT-4o and LLaVA-1.5 (7b)) in our benchmark across three types of questions: "verification" questions (supported by DSG, eg. "Is there a bench?"), "query" type questions (not supported by DSG eg. ''What is the color of the bike next to the bench?'') and "describe" type questions (not supported by DSG eg.  “Can you describe the condition of the bench?”). We tabulate these results below:
> > > >
> > > > |                  |   verify |   query |  describe |
> > > > |:-----------------|------------------:|-------------------:|-------------------:|
> > > > | LLaVA-1.5 (7b)    |    	79.2   |  **83.1**	| 	**81.9**	|
> > > > | GPT-4o       |          **79.7**	|  81.7	|	79.7	|
> > > >
> > > > Clearly, both models hallucinate across _all_ three question types, not just on verification type questions. Further, relying only on verification type questions can in this case lead to the misleading conclusion that GPT-4o is more truthful than LLaVA, whereas this is not the case consistently across question types. It is thus crucial to simulate a diverse range of questions to stress-test model hallucination, which DSG simply does not support out-of-the box. As we note previously, PROVE's technical contributions over DSG are precisely in the service of _generating, verifying, and evaluating_ such open-ended visual questions with free-form responses.
> > > >
> > > > Finally, we work through a simple example to demonstrate the limitations of DSG in image-to-text generation in the wild, and how PROVE addresses them. Consider this image from the DOCCI dataset: [test_04093.jpg](https://storage.googleapis.com/docci/thumbnails/test_04093.jpg)
> > > >
> > > > When asked the DSG-style question "Does the highway in the image have three lanes?", GPT-4o correctly responds "Yes". However, when asked "How many lanes does the highway have"?, GPT-4o incorrectly responds "The highway in the image has _four_ lanes"
> > > >
> > > > This illustrates the potential pitfalls of relying only on binary questions for hallucination evaluation -- even for what is essentially the same question, one phrasing elicits a hallucination while the other does not. By supporting a wide range of questions, we seek to surface such failure modes.

---

> > > > > ### Author Response · Authors · 2024-11-27
> > > > > **Official comment by authors**
> > > > >
> > > > > Thank you again for your thoughtful feedback. We sincerely hope that our additional clarifications have improved your evaluation of our work. Please let us know if there are any other concerns that we might be able to address.

---

### Official Review · Reviewer_GDu3 · 2024-10-30

**Soundness:** 3
**Presentation:** 3
**Contribution:** 2
**Rating:** 5
**Confidence:** 4

**Summary:**

The paper proposes a new evaluation paradigm, named PROVE, for evaluating the “helpfulness” and “trustfulness” of VLM. The evaluation is based on the newly proposed dataset, where the images are paired with LLM generated dense scene graphs using detailed descriptions and the QA pairs, together with an executable program to derive/verify the answer, are generated using LLMs. Then a “helpfulness” score (measuring the recall of entities) an “trustfulness” score (measuring the precision) are defined based on the program and the scene graph. Two limited human evaluations are provided, verifying the correctness of the eval paradigm. Multiple VLMs are evaluated, showing that these models can hardly balance the two metrics.

**Strengths:**

1. The definition of the two metrics, i.e. helpfulness and trustfulness, based on the scene graphs, is interesting.
2. The writing is clear and easy to follow.

**Weaknesses:**

1. Generalizability of the proposed evaluation paradigm is limited. The evaluation requires a dense scene graph and an executable program, which limits its usage to only the proposed dataset. The evaluation can be hardly generalized to images/questions without detailed annotations. Moreover, the evaluation’s effectiveness is bounded by the quality of the dense scene graph/detailed caption. Anything that is not in the scene graph cannot be evaluated. This is not exactly a “in-the-wild” evaluation as the paper claimed.
2. What is the advantage of the proposed method, over the claim-based evaluation method, where the model’s answer is decomposed into several claims, then LLMs verify each of the claims directly? The advantage of the latter includes that it is more flexible (does not require scene graph/program, thus can be applied to open world images), and more simple (thus is easier to apply).
3. The human study shows Pearson correlation of 0.81 for helpfulness and 0.45 for trustfulness, which is not super-high (especially for trustfulness). Any analysis on this? Another human verification can be conducted in a side-by-side manner: given 2 model responses for the same question, let human raters annotate which one is better (in terms of helpfulness and trustfulness), then use this side-by-side human rating to check the evaluation scores.

**Questions:**

See weakness.

---

> ### Author Response · Authors · 2024-11-26
> **Author response**
>
> **[Evaluation requires dense scene graph and an executable program]** This is incorrect. Our approach does not require scene graph and verification program annotations but rather generates them on-the-fly from dense image captions. As a result, our method can be readily generalized to other datasets with dense captions [A-E].
>
> **[Evaluation depends on caption quality / coverage ]** We agree and have taken several measures to ensure quality and coverage:
>
> – Quality: The DOCCI dataset, used in our benchmark, underwent a rigorous 3-stage human annotation process.
>
> – Coverage: DOCCI captions average 136 words/caption, significantly exceeding other datasets and correlating with higher image recall.
>
> – Postprocessing: As detailed in L228-245, we applied programmatic and text-based filtering to ensure QA pairs in PROVE are verifiable, unambiguous, and challenging.
>
> While elements absent from the scene graph cannot be evaluated, PROVE’s scene graphs are significantly more detailed (40.9 ± 15.2 tuples per image) than those in existing benchmarks, providing critical context for evaluating free-form VLM responses.
>
> **[Is evaluation truly “in the wild”?]** Our goal is to realistically simulate “in the wild” usage at scale. Existing efforts lack either scale (e.g., <100 examples in LLaVA-in-the-Wild, MMHalBench) or reliability (e.g., GAVIE, CIEM rely on unverified LLM-generated QA pairs). PROVE combines creative LLM-generated QA pairs with robust external verification, yielding a comprehensive and reliable evaluation testbed.
>
> **[Claim-based vs. graph-based evaluation]** While graph-based verification is a generalization of claim-based verification, we disagree that it is inherently less flexible or generalizable. We first reiterate that our method does not require pre-annotated scene graphs or verification programs but generates them dynamically, and so is as widely applicable.
>
> Next, we demonstrate how the added complexity of graph-based evaluation enhances reliability. To do so, we benchmark a claim-based LLM-as-judge evaluation strategy with LLaMA-3.1-Instruct (8b). We provide the model with the ground truth caption as context and prompt it to break down responses into atomic claims and score their helpfulness and truthfulness. We repeat the evaluation process for four models (LLaVA-1.5, LLaVA-Next, GPT-4o-mini, GPT-4o,  L376-405) and measure Pearson correlation with human judgments:
>
>
> |                     |   helpfulness	  |  truthfulness |
> |:-----------------|------------------:|-------------------:|
> | Claim-based LLM | 0.41	          |	0.20		|
> | Ours             |  **0.81** 	|	**0.45**|
>
> As seen, while the observed correlation scores are still objectively far-from-perfect, they are relatively far superior to existing LLM-as-judge approaches.
>
> **[Analyzing modest tscore correlation]** As recommended, we investigate the relatively low correlation observed for tscore and human judgement of truthfulness. For each QA pair, we compare human-assigned truthfulness scores to our proposed tscore (computed via image-text and text-text matching, see Eq. 2 of our paper). On average, tscore overestimates truthfulness by 11%, primarily due to the image-text matching; removing it causes tscore to _underestimate_ truthfulness by 15%. However, removing image-text matching also reduces the Pearson correlation with human scores from 0.45 to 0.39, highlighting the complementary strengths of both components.
>
> We sincerely hope that these clarifications persuade you to reconsider your evaluation, and would be happy to address any lingering concerns.
>
> [A] Krause, Jonathan, et al. "A hierarchical approach for generating descriptive image paragraphs." Proceedings of the IEEE conference on computer vision and pattern recognition. 2017.
>
> [B] Pont-Tuset, Jordi, et al. "Connecting vision and language with localized narratives." Computer Vision–ECCV 2020: 16th European Conference, Glasgow, UK, August 23–28, 2020, Proceedings, Part V 16. Springer International Publishing, 2020.
>
> [C] Urbanek, Jack, et al. "A picture is worth more than 77 text tokens: Evaluating clip-style models on dense captions." Proceedings of the IEEE/CVF Conference on Computer Vision and Pattern Recognition. 2024.
>
> [D] Garg, Roopal, et al. "ImageInWords: Unlocking Hyper-Detailed Image Descriptions." arXiv preprint arXiv:2405.02793 (2024).
>
> [E] Awadalla, Anas, et al. "BLIP3-KALE: Knowledge Augmented Large-Scale Dense Captions." arXiv preprint arXiv:2411.07461 (2024)

---

> > ### Author Response · Authors · 2024-11-27
> > **Official comment by authors**
> >
> > Thank you again for your thoughtful feedback. We sincerely hope that our response above has adequately addressed your concerns and improved your evaluation of our work. Please let us know if there is anything else that we might be able to clarify.

---

### Official Review · Reviewer_ubc1 · 2024-11-03

**Soundness:** 2
**Presentation:** 2
**Contribution:** 3
**Rating:** 5
**Confidence:** 4

**Summary:**

This paper proposes a new benchmark which the authors constructed by turning detailed captions into scene graph representations, and generating QA pairs as well as the corresponding verification programs based on the scene graphs. It also proposes a programmatic evaluation of the VLMs’ responses on both helpfulness and truthfulness by comparing the predicted scene graphs and ground-truth scene graphs.

**Strengths:**

- The paper is well-motivated and tackles an important research problem in VLMs evaluation. The inclusion of truthfulness in addition to helpfulness is thoughtful and often neglected.
- The paper is generally well-written with clear definitions of the helpfulness and truthfulness metrics, and helpful illustrations like figure 4.
- The evaluation covers a broad range of models.
- The authors perform multiple data filtering steps to ensure the correctness of the programs and high quality of the QA pairs.

**Weaknesses:**

- The reviewer is mostly concerned about the use of models in multiple parts of the dataset generation, filtering, and evaluation pipeline, especially in extracting the scene graphs from captions.
- For example, the scene graphs are not guaranteed to be completely accurate, as they are automatically extracted from the captions in the DOCCI dataset by an LLM without any human verification or filtering.
- Similarly, as the authors mentioned, the sentence BERT model and visual entailment model OFA are used for metrics calculation, which means the evaluation accuracy is limited by these models’ accuracies, making this a much less rigorous evaluation benchmark. Have the authors analyzed how errors in these models might propagate through the evaluation?

**Questions:**

Questions on the human study:
- Are human ratings on helpfulness continuous or discrete?
- The correlation score with the helpfulness score is 0.54 and quite low – can the authors provide insights into why this is the case? Is there anything humans pay more attention to for helpfulness that the hscore doesn’t capture?

---

> ### Author Response · Authors · 2024-11-26
> **Author response**
>
> [**Concern about use of models**] This is a valid concern towards which we have devoted considerable thought and effort.
>
> **Evaluating LLM-generated scene graph.**  The scene graphs in the DOCCI dataset are generated using Palm-2 (340B) following the DSG [A] method. A human study in DSG shows that these scene graphs achieve 92.2% precision and 100% recall on TIFA160, demonstrating high quality.
>
> **Evaluating model-based metric computation.** Next, we study potential errors in text-text (via Sentence-BERT) and image-text (via OFA) matching and how those may propagate. We focus on truthfulness evaluation where we observe lower correlation with human judgment. For each QA pair, we compare human-assigned truthfulness scores to our proposed tscore (computed via image-text and text-text matching, see Eq. 2 of our paper). On average, tscore _overestimates_ truthfulness by 11%, primarily due to OFA; removing image-text matching causes tscore to _underestimate_ truthfulness by 15%. However, removing image-text matching also reduces the Pearson correlation with human scores from 0.45 to 0.39, highlighting the complementary strengths of both components.
>
> To demonstrate robustness, we visualize the evaluation pipeline for various models and examples here: https://prove-explorer-anon.netlify.app.
>
> Further, we note that our reliance on external models is limited to well-defined subtasks (e.g., tuple extraction, text matching) for which these models are specialized and continually improving. Unlike LLM-as-judge scoring, our method provides an interpretable and auditable trace of how responses are scored, which mitigates errors and enhances transparency.
>
> In summary, while external models can introduce errors, we believe that these are largely mitigated by our robust approach. The near-perfect human ratings for relevance and correctness of QA pairs in our benchmark (L369-374) further validate this.
>
> [**Human study clarification**] Human ratings for hscore and tscore are discrete (0/1 and 0/0.5/1.0, respectively, as noted in L377). We discovered a bug in our correlation computation after fixing which observe Pearson correlation scores of 0.81 (hscore) and 0.45 (tscore).
>
> To contextualize these scores, we benchmarked an LLM-as-judge baseline (LLaMA-3.1-Instruct, 8b), following the MMHal-Bench protocol. Using the same prompts and procedures (L376-405), we evaluated responses from four models (LLaVA-1.5, LLaVA-Next, GPT-4o-mini, GPT-4o) for helpfulness and truthfulness, correlating these scores with human judgments:
>
>
> |                     |   helpfulness	  |  truthfulness |
> |:-----------------|------------------:|-------------------:|
> | LLM-as-judge | 0.41	          |	0.20		|
> | Ours             |  **0.81** 	|	**0.45**|
>
>
> As seen, our proposed metrics correlate _substantially_ better with human judgment compared to a pure LLM-as-judge approach.
>
> We sincerely hope that these clarifications persuade you to reconsider your evaluation, and would be happy to address any lingering concerns.

---

> > ### Comment · Reviewer_ubc1 · 2024-11-26
> >
> > Thank the authors for their responses, especially on the error propagation analysis!
> >
> > While the authors mentioned that they'd discuss errors in both text-text (via sentence BERT) and image-text matching, it seems like the numbers for text-text matching errors are missing? Also, it seems like the image-text matching errors are quite impactful -- do the authors have ideas/plans about mitigating these errors?
> >
> > The comparison to LLM-as-judge is quite interesting and can make a convincing argument for this work -- the reviewer wonders: what do the correlation scores look like if the LLM judge were GPT-4o/GPT-4o mini instead of LLaMA3.1?

---

> > > ### Author Response · Authors · 2024-11-26
> > > **Author response**
> > >
> > > Thank you for considering our response and for the additional thoughtful feedback.
> > >
> > > **[Stronger judge LLM]** As suggested, we experiment with a stronger LLM judge instead of LLaMA-3.1 Results are below:
> > >
> > > |  Judge                  |   helpfulness	  |  truthfulness |
> > > |:-----------------|------------------:|-------------------:|
> > > | LLaMA-3.1 (8b) | 0.41	          |	0.20		|
> > > | GPT-4o-mini  |	     0.58     |	0.30		|
> > > | Ours             |  **0.81** 	|	**0.45**|
> > >
> > > As seen, the stronger judge model improve correlation but still fall short of our proposed method. In Figure 11 of the revised appendix we include real examples from existing hallucination benchmarks that qualitatively highlight the limitations of LLM judges in open-ended response evaluation.
> > >
> > > **[Evaluating text-text matching]** Our hscore calculation, based solely on text-text matching, is well-suited for this task. Using human assessments of response helpfulness as the gold standard, we find that hscore underestimates helpfulness by 10.9% on average. This relatively small deviation indicates strong overall alignment with human judgment, with high precision (few spurious matches) but imperfect recall (some true matches missed) which translate to the high overall Pearson correlation that we observe (0.81).
> > >
> > > For qualitative evidence, we again refer the reviewer to https://prove-explorer-anon.netlify.app.
> > >
> > > **[Mitigating image-text matching errors]** We could potentially improve image-text matching via:
> > >
> > > – Region-Level Matching: Measuring region-level image-text entailment could provide more granular and robust estimates.
> > >
> > > – Enriched Scene Graphs: Incorporating additional human/machine annotations (e.g., bounding boxes, semantic segmentation masks) and fine-tuning the visual entailment model could enhance reliability.
> > >
> > > Ultimately, we hope that iterative enrichment of the scene graph may achieve sufficiently high image coverage—leading to a “Platonic” representation [A]—where text-based matching alone suffices.
> > >
> > > We hope this addresses the reviewer’s remaining concerns and are happy to provide further clarification if needed.
> > >
> > > [A] Huh, Minyoung, et al. "The platonic representation hypothesis." ICML 2024.

---

> > > > ### Comment · Reviewer_ubc1 · 2024-11-27
> > > >
> > > > The reviewer thanks the authors for their responses!
> > > > While the reviewer thinks this submission is close to acceptance, they believe that this work would be stronger if the authors improved on the correlation between truthfulness and human judgement as it is currently 0.45 and still very low (even though it is higher than LLM judge). As the authors already identified the errors and mentioned ways to mitigate them, the reviewer believes that they'd be able to enhance the evaluation pipeline by reducing these errors. The reviewer would like to keep their score as it is.

---

> > > > > ### Author Response · Authors · 2024-12-02
> > > > > **Author response**
> > > > >
> > > > > Thank you for the additional feedback.  Following the reviewer's suggestion, we implement one of our proposed strategies. Specifically, we update our image-text matching algorithm from OFA to Grounding DINO [A] -- a state-of-the-art open-set object detector, and obtain tuple grounding scores predicted by this model (rather than OFA) as our image-text entailment score. We then recompute tscore and measure Pearson correlation with human judgement. We find this gives us a higher correlation of **0.567** versus **0.451** with OFA and **0.30** with a GPT-4o-mini LLM as judge.
> > > > >
> > > > > We acknowledge that even this improved tscore leaves room for improvement, but emphasize that this attests to the difficulty of reliably evaluating hallucinations in free-form responses, which neccesitates enumerating and visually verifying _each_ claim made in the response. In this work we take a step towards this challenging goal by proposing a ground-up redesign of the VLM evaluation stack favoring scene-graph based programmatic verification and evaluation over black-box LLM-as-judge scoring, and demonstrate its improved flexibility (jointly evaluating helpfulness and truthfulness of open-ended responses) and reliability (higher correlation with human judgement).
> > > > >
> > > > > As the discussion period draws to a close, we hope that our additional experiments and clarifications further improves the reviewer's opinion of our work. We are sincerely grateful for their thoughtful engagement and feedback on our work.
> > > > >
> > > > > [A] Liu, Shilong, et al. "Grounding dino: Marrying dino with grounded pre-training for open-set object detection." European Conference on Computer Vision. Springer, Cham, 2025.

---

### Official Review · Reviewer_wMz9 · 2024-11-04

**Soundness:** 2
**Presentation:** 2
**Contribution:** 2
**Rating:** 5
**Confidence:** 4

**Summary:**

Programmatic VLM Evaluation (PROVE) introduces a novel benchmark for assessing VLMs. Normally, we evaluate image captioning with the generated caption and the gold caption as two whole paragraphs. Building upon DOCCI, which is a new dataset that came out this year, PROVE collects a robust set of 10.5k visually grounded question-answer (QA) pairs by using a detailed scene-graph approach that evaluates image captions compositionally. It provides a programmatic evaluation strategy that measures both the helpfulness and truthfulness of a free-form model responses.

**Strengths:**

I do like how this method could evaluate VLM compositionally with a prepared set of programs, instead of all in a whole with a LLM.
The design of helpfulness and truthfulness is interesting. It is interesting to find a way to evaluate hallucination in VLMs.

**Weaknesses:**

1. The presentation is poor. It took me a while to finally realize that this paper presents a way to evaluate image-captioning, through breaking the captioning task into VQA tasks, and the answers are evaluated by a program generated by GPT based on gold scene graph, instead of evaluated by GPT based on the gold caption.

2. The results in Table1 and example outputs in Figure 5 are very confusing. Why are the performance of all models look similar? From my personal experience, GPT-4o should be better than other models, especially the open-source models by a lot. But they seem to have same performance as in Table1. From the original DOCCI paper, different models also score very differently. From Figure 5 in the first example's top question, I don't understand why the hscore for GPT-4o and LLaVA are both so high -- none of them gave the correct answer. In that same example's bottom question, I don't understand why the hscore for LLaVA is so low, given that it answers the question perfectly.

3.  All the questions are generated by LLM. This could potentially only include easy questions, and might explain why the performance in Table 1 are all similar.

4. The Oracle setting result is way too low -- only 4 % higher than all other models. Isn't the Oracle setting the same setting you applied when generating the QA dataset? Shouldn't this setting then achieve 100% in accuracy?

5. The average cosine similarity score in (1) and (2) is not very convincing. From Figure 5 in the first example's top question, I don't understand why the hscore for GPT-4o and LLaVA are both so high -- none of them gave the correct answer. In that same example's bottom question, I don't understand why the hscore for LLaVA is so low, given that it answers the question perfectly.


===

Thanks for the clarifications. I've adjusted my rating given the additional scores and explanations.

**Questions:**

See Weakness

---

> ### Author Response · Authors · 2024-11-20
> **Author response (1/2)**
>
> **["this paper presents a way to evaluate image-captioning"]** We believe that there has been a misunderstanding: PROVE is a benchmark to evaluate **vision-language models (VLMs)** in the wild, rather than image captioning models alone. As reviewer ubc1 succinctly summarizes:
>
> *This paper proposes a new benchmark which the authors constructed by turning detailed captions into scene graph representations, and generating QA pairs as well as the corresponding verification programs based on the scene graphs. It also proposes a programmatic evaluation of the VLMs’ responses on both helpfulness and truthfulness by comparing the predicted scene graphs and ground-truth scene graphs.*
>
> Concretely, we use human-generated image captions to construct a ''gold scene graph'' which serves two complementary purposes:
>
> 1) **Dataset verification**. For each image, we programmatically verify each question-answer pair generated by an LLM (GPT-4o) from its gold scene graph; specifically, that the question is even answerable for the image, and that the generated answer is accurate.
>
> 2) **VLM Evaluation**. At test time, we evaluate both the helpfulness and truthfulness of each free-form VLM response by comparing the response’s scene-graph representation with the gold scene graph. Such scoring provides a more reliable and interpretable scoring rationale than a pure ''LLM as judge'' approach.
>
> We will rephrase the text to improve clarity, and welcome suggestions towards the same.
>
> **["Why does the performance across models in Table 1 look similar? .. GPT-4o should be better than other models"]** To clarify, performance across methods is *not* identical: across surveyed methods, the largest performance gap is 6.5% (absolute) for hscore and 5.75% (absolute) for tscore. However, due to the fact that the two metrics tend to be negatively correlated (see Figure 4), the maximum _average_ performance gap is relatively small (3.3% absolute).
>
> Further, GPT-4o does indeed score highest on the (comprehension and reasoning focused) hscore measure, matching expectations. However, we find that it also tends to hallucinate more than some other methods, resulting in a relatively low tscore. It is precisely this ability to jointly evaluate helpfulness and truthfulness, which existing benchmarks lack, that motivates our work.
>
> However, we agree that our continuous cosine similarity based metric is not conducive to _fine-grained_ discrimination between methods. To address this, we propose a discrete version based on thresholding, defined as:
>
> $$
> \text{hscore}^{\theta} (\hat{A})=\frac{\sum_{t \in g(A)-g(Q)} \mathbb{I}\big[\max_{t' \in g(\hat{A})} \text{sim}(t, t') > \theta\big]}{|g(A)-g(Q)|}
> $$
>
> $$
> \text{tscore}^{\theta} (\hat{A}) = \frac{\sum_{t' \in g(\hat{A})}  \mathbb{I}\big[\text{max}\big(\max_{t \in g(C)}(\text{sim}(t', t), p(I \models t')\big) > \theta\big]}{|g(\hat{A})|}
> $$
>
> Recomputing (a subset of) methods in Table 1 with these updated measures gives us:
>
> |                  |   $\text{hscore}^{0.75}$  | $\text{tscore}^{0.75}$ |   $\text{average}^{0.75}$ |
> |:-----------------|------------------:|-------------------:|--------------:|
> | Qwen2-VL           |             45.7  |              82.79 |         64.25 |
> | Phi-3.5-Vision            |             51.74 |              86.13 |         68.93 |
> | LLaVA-1.5 (7b)            |             50.5  |              **87.33** |         68.91 |
> | InternVL2 (8b)       |             53.87 |              84.61 |         69.24 |
> | Pixtral (12b)          |             52.13 |              86.79 |         69.46 |
> | InternVL2 (26b)      |             54.64 |              82.24 |         68.44 |
> | Gemini-1.5-Flash |             50.65 |              84.04 |         67.35 |
> | GPT-4o  |             **57.89** |              85.74 |         **71.81** |
>
> As seen, thresholding leads to a lower hscore and higher tscore across the board, more than doubling the maximum gap in average performance between methods to 7.5% (absolute), _without_ changing the overall ranking -- GPT-4o still scores highest on helpfulness and LLaVA-1.5 (7b) on truthfulness We will be happy to revise our paper with these fine-grained results.
>
> **Fig. 5 correction**. Thanks for bringing this to our notice. LLaVA-1.5 (7b)’s actual response to Q2 for the first image is ''The bricks at the bottom of the wall are _white_'', which results in the low score -- we will revise the paper to fix this. For Q1, GPT-4o’s answer ''BISTER'' is more similar to the ground truth answer ''BUSIER'' in Sentence BERT embedding space than LLaVA-1.5-7b’s (''MISTHI''), which is why it obtains a higher hscore.

---

> > ### Author Response · Authors · 2024-11-20
> > **Author response (2/2)**
> >
> > **[Are the LLM-generated questions easy?]** On the contrary, we explicitly prompt the LLM to generate challenging (but still verifiable) questions, and on manual review find this to be the case. We include a random subset of 500 examples from our benchmark for this reviewer’s consideration at this link: [https://prove-explorer-anon.netlify.app](https://prove-explorer-anon.netlify.app)
> >
> > Further, to illustrate this quantitatively, we report performance below by question type over ''describe'' type questions (eg. “Can you describe the condition of the bench?”, ~4.1k queries in total) and ''query'' type questions (eg. ''What is the color of the bike next to the bench?'', 6.2k queries in total).
> >
> > |                  |   query hscore$^{0.75}$ |   query tscore$^{0.75}$ |  describe hscore$^{0.75}$ |   describe tscore$^{0.75}$  |
> > |:-----------------|------------------:|-------------------:|--------------:|--------------:|
> > | LLaVA-1.5 (7b)    |    	50.95    |  **87.55**	| 	49.70	| **87.14** |
> > | GPT-4o       |          **58.08**	|   86.54	|	**57.55**	| 84.53 |
> >
> > Across both question types, a frontier model like GPT-4o strongly outperforms LLaVA-1.5 (7b) on hscore$^{0.75}$ (+8% absolute each). However, while GPT-4o only lags LLaVA-1.5 (7b) in tscore$^{0.75}$ (-1% absolute) for more targeted ''query'' type questions, it does so more significantly (-2.61% absolute) for descriptive questions, wherein the open-ended nature of the questions leads to a higher degree of hallucination.
> >
> > **[Why is oracle performance not 100%?]** The ''oracle'' model we benchmark is a blind LLM (LLaMA-3.1-Instruct (8B)) that has additional access to the ground truth image caption (L305, perhaps a more suitable name would be  ''Blind LLM with caption''? ) with the intention of obtaining a loose upper bound on performance. Unlike the LLM that we use to generate QA pairs in PROVE however, it does _not_ have access to the image scene graph, or a mechanism to programmatically verify its responses. As a result, while it does achieve a relatively high average$^{0.75}$ (+7.7% absolute over best VLM), it is still susceptible to hallucination. We will clarify this, thanks!
> >
> > We sincerely hope that these clarifications persuade you to reconsider your evaluation, and would be happy to address any lingering concerns.

---

> > > ### Author Response · Authors · 2024-11-25
> > > **Official comment by authors**
> > >
> > > We sincerely appreciate your time and valuable feedback on our work. As we are in the discussion phase, we would welcome further dialogue to ensure our responses address your concerns and to explore any additional questions or points you may have.

---

> > > > ### Author Response · Authors · 2024-11-27
> > > > **Official comment by authors**
> > > >
> > > > Thank you again for your thoughtful feedback. We sincerely hope that our response above has adequately addressed your concerns and improved your evaluation of our work. Please let us know if there is anything else that we might be able to clarify.

---

### Meta-Review · Area_Chair_A7d7 · 2024-12-20

**Metareview:**

The authors present Programmatic VLM Evaluation (PROVE), which introduces a new benchmark for assessing VLMs. The AC appreciates the authors' efforts and improvements. However, explicit qualitative and quantitative comparisons between DSG and PROVE are essential to establish the necessity and effectiveness of the proposed framework. The low correlation (0.45) between truthfulness and human judgment also weakens the evaluation. Addressing these issues would significantly strengthen the submission. The AC shares the lack of enthusiasm with all four reviewers, but would encourage the authors to revise and improve.

**Additional Comments On Reviewer Discussion:**

Some discussions happened, AC appreciates them.

---

### Decision · Program_Chairs · 2025-01-22

Reject